# Researchers working from home: Benefits and challenges

**Balazs Aczel** [1]*, **Marton Kovacs**[1,2], **Tanja van der Lippe**[3], **Barnabas Szaszi**[1]

**1** Institute of Psychology, ELTE Eotvos Lorand University, Budapest, Hungary, **2** Doctoral School of Psychology, ELTE Eotvos Lorand University, Budapest, Hungary, **3** Department of Sociology, Utrecht University, Utrecht, The Netherlands

* aczel.balazs@ppk.elte.hu

**Data Availability Statement:** All research materials, the collected raw and processed anonymous data, just as well the code for data management and statistical analyses are publicly

## Abstract

The flexibility allowed by the mobilization of technology disintegrated the traditional work-life boundary for most professionals. Whether working from home is the key or impediment to academics' efficiency and work-life balance became a daunting question for both scientists and their employers. The recent pandemic brought into focus the merits and challenges of working from home on a level of personal experience. Using a convenient sampling, we surveyed 704 academics while working from home and found that the pandemic lockdown decreased the work efficiency for almost half of the researchers but around a quarter of them were more efficient during this time compared to the time before. Based on the gathered personal experience, 70% of the researchers think that in the future they would be similarly or more efficient than before if they could spend more of their work-time at home. They indicated that in the office they are better at sharing thoughts with colleagues, keeping in touch with their team, and collecting data, whereas at home they are better at working on their manuscript, reading the literature, and analyzing their data. Taking well-being also into account, 66% of them would find it ideal to work more from home in the future than they did before the lockdown. These results draw attention to how working from home is becoming a major element of researchers' life and that we have to learn more about its influencer factors and coping tactics in order to optimize its arrangements.

## Introduction

Fleeing from the Great Plague that reached Cambridge in 1665, Newton retreated to his countryside home where he continued working for the next year and a half. During this time, he developed his theories on calculus, optics, and the law of gravitation—fundamentally changing the path of science for centuries. Newton himself described this period as the most productive time of his life [1]. Is working from home indeed the key to efficiency for scientists also in modern times? A solution for working without disturbance by colleagues and being able to manage a work-life balance? What personal and professional factors influence the relation between productivity and working from home? These are the main questions that the present

shared on the OSF page of the project: OSF: https://osf.io/v97fy/.

**Funding:** TVL's contribution is part of the research program Sustainable Cooperation – Roadmaps to Resilient Societies (SCOOP). She is grateful to the Netherlands Organization for Scientific Research (NWO) and the Dutch Ministry of Education, Culture and Science (OCW) for their support in the context of its 2017 Gravitation Program (grant number 024.003.025).

**Competing interests:** The authors have declared that no competing interests exist.

paper aims to tackle. The Covid-19 pandemic provides a unique opportunity to analyze the implications of working from home in great detail.

Working away from the traditional office is increasingly an option in today's world. The phenomenon has been studied under numerous, partially overlapping terms, such as telecommuting, telework, virtual office, remote work, location independent working, home office. In this paper, we will use 'working from home' (WFH), a term that typically covers working from any location other than the dedicated area provided by the employer.

The practice of WFH and its effect on job efficiency and well-being are reasonably well explored outside of academia [2, 3]. Internet access and the increase of personal IT infrastructure made WFH a growing trend throughout the last decades [4]. In 2015, over 12% of EU workers [5] and near one-quarter of US employees [6] worked at least partly from home. A recent survey conducted among 27,500 millennials and Gen Z-s indicated that their majority would like to work remotely more frequently [7]. The literature suggests that people working from home need flexibility for different reasons. Home-working is a typical solution for those who need to look after dependent children [8] but many employees just seek a better work-life balance [7] and the comfort of an alternative work environment [9].

Non-academic areas report work-efficiency benefits for WFH but they also show some downsides of this arrangement. A good example is the broad-scale experiment in which call center employees were randomly assigned to work from home or in the office for nine months [10]. A 13% work performance increase was found in the working from home group. These workers also reported improved work satisfaction. Still, after the experiment, 50% of them preferred to go back to the office mainly because of feeling isolated at home.

Home-working has several straightforward positive aspects, such as not having to commute, easier management of household responsibilities [11] and family demands [12], along with increased autonomy over time use [13, 14], and fewer interruptions [15, 16]. Personal comfort is often listed as an advantage of the home environment [e.g., 15], though setting up a home office comes with physical and infrastructural demands [17]. People working from home consistently report greater job motivation and satisfaction [4, 11, 18, 19] which is probably due to the greater work-related control and work-life flexibility [20]. A longitudinal nationally representative sample of 30,000 households in the UK revealed that homeworking is positively related with leisure time satisfaction [21], suggesting that people working from home can allocate more time for leisure activities.

Often-mentioned negative aspects of WFH include being disconnected from co-workers, experiencing isolation due to the physical and social distance to team members [22, 23]. Also, home-working employees reported more difficulties with switching off and they worked beyond their formal working hours [4]. Working from home is especially difficult for those with small children [24], but intrusion from other family members, neighbours, and friends were also found to be major challenges of WFH [e.g., 17]. Moreover, being away from the office may also create a lack of visibility and increases teleworkers' fear that being out of sight limits opportunities for promotion, rewards, and positive performance reviews [25].

Importantly, increased freedom imposes higher demands on workers to control not just the environment, but themselves too. WFH comes with the need to develop work-life boundary control tactics [26] and to be skilled at self-discipline, self-motivation, and good time management [27]. Increased flexibility can easily lead to multitasking and work-family role blurring [28]. Table 1 provides non-comprehensive lists of mostly positive and mostly negative consequences of WFH, based on the literature reviewed here.

Compared to the private sector, our knowledge is scarce about how academics experience working from home. Researchers in higher education institutes work in very similar arrangements. Typically, they are expected to personally attend their workplace, if not for teaching or

**Table 1. Positive and negative consequences of WFH.**

| Mostly positive | Mostly negative |
| --- | --- |
| Less commuting | Isolation from colleagues |
| More control over time | Less defined work-life boundaries |
| More autonomy | Higher need for self-discipline |
| Less office-related distractions | Reliance on private infrastructure |
| More comfortable environment | Communication difficulties with colleagues |
| More flexibility with domestic tasks | |

supervision, then for meetings or to confer with colleagues. In the remaining worktime, they work in their lab or, if allowed, they may choose to do some of their tasks remotely. Along with the benefits on productivity when working from home, academics have already experienced some of its drawbacks at the start of the popularity of personal computers. As Snizek observed in the '80s, "(f)aculty who work long hours at home using their microcomputers indicate feelings of isolation and often lament the loss of collegial feedback and reinforcement" [page 622, 29].

Until now, the academics whose WFH experience had been given attention were mostly those participating in online distance education [e.g., 30, 31]. They experienced increased autonomy, flexibility in workday schedule, the elimination of unwanted distractions [32], along with high levels of work productivity and satisfaction [33], but they also observed inadequate communication and the lack of opportunities for skill development [34]. The Covid-19 pandemic provided an opportunity to study the WFH experience of a greater spectrum of academics, since at one point most of them had to do all their work from home.

We have only fragmented knowledge about the moderators of WFH success. We know that control over time is limited by the domestic tasks one has while working from home. The view that women's work is more influenced by family obligations than men's is consistently shown in the literature [e.g., 35–37]. Sullivan and Lewis [38] argued that women who work from home are able to fulfil their domestic role better and manage their family duties more to their satisfaction, but that comes at the expense of higher perceived work–family conflict [see also 39]. Not surprisingly, during the COVID-19 pandemic, female scientists suffered a greater disruption than men in their academic productivity and time spent on research, most likely due to demands of childcare [40, 41].

In summary, until recently, the effect of WFH on academics' life and productivity received limited attention. However, during the recent pandemic lockdown, scientists, on an unprecedented scale, had to find solutions to continue their research from home. The situation unavoidably brought into focus the merits and challenges of WFH on a level of personal experience. Institutions were compelled to support WFH arrangements by adequate regulations, services, and infrastructure. Some researchers and institutions might have found benefits in the new arrangements and may wish to continue WFH in some form; for others WFH brought disproportionately larger challenges. The present study aims to facilitate the systematic exploration and support of researchers' efficiency and work-life balance when working from home.

## Materials and methods

Our study procedure and analysis plan were preregistered at https://osf.io/jg5bz (all deviations from the plan are listed in S1 File). The survey included questions on research work efficiency, work-life balance, demographics, professional and personal background information. The study protocol has been approved by the Institutional Review Board from Eotvos Lorand

University, Hungary (approval number: 2020/131). The Transparency Report of the study, the complete text of the questionnaire items and the instructions are shared at our OSF repository: https://osf.io/v97fy/.

## Sampling

As the objective of this study was to gain insight about researchers' experience of WFH, we aimed to increase the size and diversity of our sample rather than ascertaining the representativeness of our sample. Therefore, we distributed our online survey link among researchers in professional newsletters, university mailing lists, on social media, and by sending group-emails to authors (additional details about sampling are in S1 File). As a result of the nature of our sampling strategy, it is not known how many researchers have seen our participation request. Additionally, we did not collect the country of residence of the respondents. Responses analyzed in this study were collected between 2020-04-24 and 2020-07-13. Overall, 858 individuals started the survey and 154 were excluded because they did not continue the survey beyond the first question. As a result, 704 respondents were included in the analysis.

## Procedure

We sent the questionnaire individually to each of the respondents through the Qualtrics Mailer service. Written informed consent and access to the preregistration of the research was provided to every respondent before starting the survey. Then, respondents who agreed to participate in the study could fill out the questionnaire. To encourage participation, we offered that upon completion they can enter a lottery to win a 100 USD voucher.

## Materials

This is a general description of the survey items. The full survey with the display logic and exact phrasing of the items is transported from Qualtrics and uploaded to the projects' OSF page: https://osf.io/8ze2g/.

**Efficiency of research work.** The respondents were asked to compare the efficiency of their research work during the lockdown to their work before the lockdown. They were also asked to use their present and previous experience to indicate whether working more from home in the future would change the efficiency of their research work compared to the time before the lockdown. For both questions, they could choose among three options: "less efficient"; "more efficient", and "similarly efficient".

**Comparing working from home to working in the office.** Participants were asked to compare working from home to working from the office. For this question they could indicate their preference on a 7-point dimension (1: At home; 7: In the office), along 15 efficiency or well-being related aspects of research work (e.g., working on the manuscript, maintaining work-life balance). These aspects were collected in a pilot study conducted with 55 researchers who were asked to indicate in free text responses the areas in which their work benefits/suffers when working from home. More details of the pilot study are provided in S1 File.

**Actual and ideal time spent working from home.** To study the actual and ideal time spent working from home, researcher were asked to indicate on a 0–100% scale (1) what percentage of their work time they spent working from home before the pandemic and (2) how much would be ideal for them working from home in the future concerning both research efficiency and work-life balance.

**Feasibility of working more from home.** With simple Yes/No options, we asked the respondents to indicate whether they think that working more from home would be feasible

considering all their other duties (education, administration, etc.) and the given circumstances at home (infrastructure, level of disturbance).

**Background information.** Background questions were asked by providing preset lists concerning their academic position (e.g., full professor), area of research (e.g., social sciences), type of workplace (e.g., purely research institute), gender, age group, living situation (e.g., single-parent with non-adult child(ren)), and the age and the number of their children.

The respondents were also asked to select one of the offered options to indicate: whether or not they worked more from home during the coronavirus lockdown than before; whether it is possible for them to collect data remotely; whether they have education duties at work; if their research requires intensive team-work; whether their home office is fully equipped; whether their partner was also working from home during the pandemic; how far their office is from home; whether they had to do home-schooling during the pandemic; whether there was someone else looking after their child(ren) during their work from home in lockdown. When the question did not apply to them, they could select the 'NA' option as well.

## Data preprocessing and analyses

All the data preprocessing and analyses were conducted in R [42], with the use of the tidyverse packages [43]. Before the analysis of the survey responses, we read all the free-text comments to ascertain that they do not contain personal information and they are in line with the respondent's answers. We found that for 5 items the respondents' comments contradicted their survey choices (e.g., whether they have children), therefore, we excluded the responses of the corresponding items from further analyses (see S1 File). Following the preregistration, we only conducted descriptive statistics of the survey results.

## Results

### Background information

The summary of the key demographic information of the 704 complete responses is presented in Table 2. A full summary of all the collected background information of the respondents are available in S1 File.

### Efficiency of research work

The results showed that 94% (n = 662) of the surveyed researchers worked more from home during the COVID-19 lockdown compared to the time before. Of these researchers, 47%

**Table 2. Number and proportion of respondents in each demographic category.**

| Background information question | Subgroup | Number of responses | Proportion of the subgroup |
| --- | --- | --- | --- |
| Gender | Female | 356 | 50.57 |
| Gender | Male | 338 | 48.01 |
| Gender | Prefer not to say | 9 | 1.28 |
| Gender | Other | 1 | 0.14 |
| Academic position | full professor | 209 | 29.69 |
| Academic position | associate professor | 172 | 24.43 |
| Academic position | assistant professor | 126 | 17.90 |
| Academic position | PhD student | 72 | 10.23 |
| Academic position | postdoc | 72 | 10.23 |
| Academic position | non-academic researcher | 38 | 5.40 |
| Academic position | research assistant | 14 | 1.99 |
| Academic position | not applicable | 1 | 0.14 |

found that due to working more from home their research became, in general, less efficient, 23% found it more efficient, and 30% found no difference compared to working before the lockdown. Within this database, we also explored the effect of the lockdown on the efficiency of people living with children (n = 290). Here, we found that 58% of them experienced that due to working more from home their research became, in general, less efficient, 20% found it more efficient, and 22% found no difference compared to working before the lockdown. Of those researchers who live with children, we found that 71% of the 21 single parents and 57% of the 269 partnered parents found working less efficient when working from home compared to the time before the lockdown.

When asking about how working more from home would affect the efficiency of their research after the lockdown, of those who have not already been working from home full time (n = 684), 29% assumed that it could make their research, in general, less efficient, 29% said that it would be more efficient, and 41% assumed no difference compared to the time before the lockdown (Fig 1).

Focusing on the efficiency of the subgroup of people who live with children (n = 295), we found that for 32% their research work would be less efficient, for 30% it would be no different, and for 38% it would be more efficient to work from home after the lockdown, compared to the time before the lockdown.

## Comparing working from home to working in the office

When comparing working from home to working in the office in general, people found that they can better achieve certain aspects of the research in one place than the other. They indicated that in the office they are better at sharing thoughts with colleagues, keeping in touch with their team, and collecting data, whereas at home they are better at working on their manuscript, reading the literature, and analyzing their data (Fig 2).

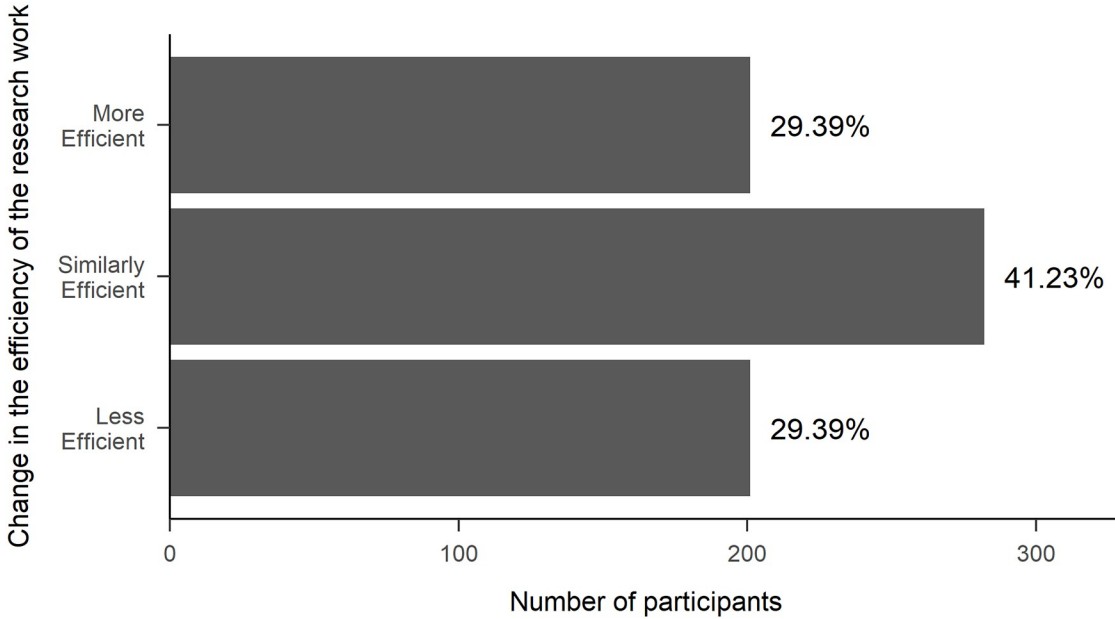

**Fig 1. Percentages of the responses (*N* = 684) given to the three answer options when asked how working more from home would affect the efficiency of their research after the lockdown.**

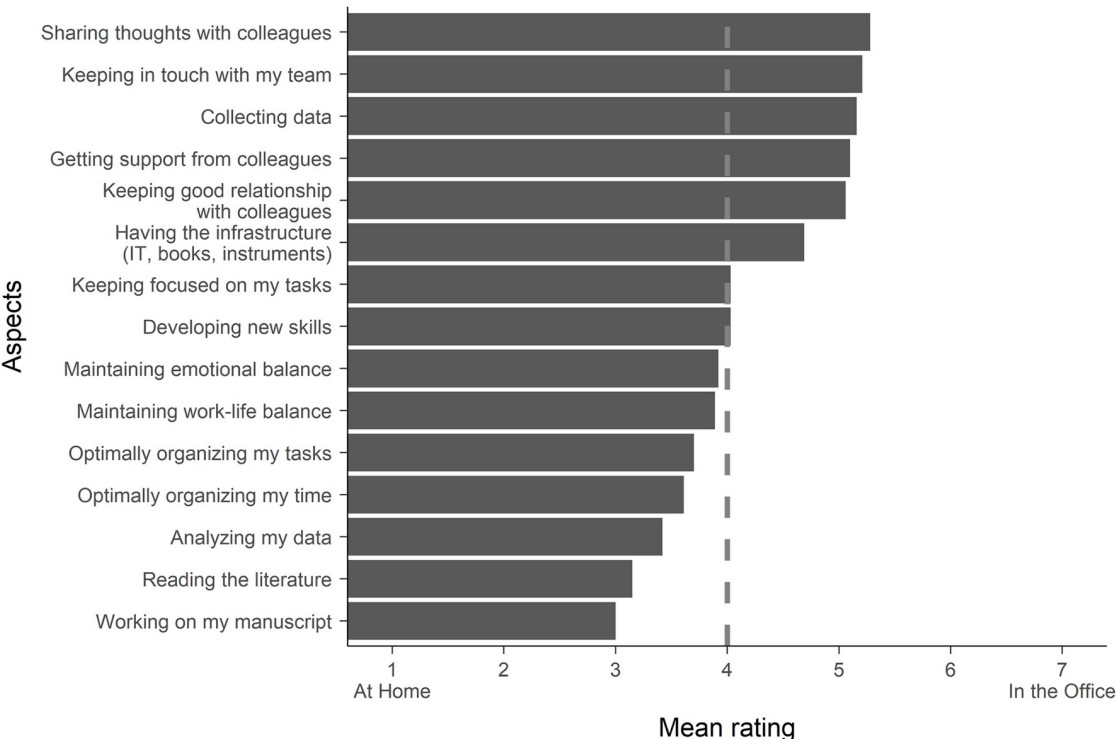

**Fig 2. The comparison ($N = 703$) of working at home and in the office concerning how the different aspects of research and work-life balance can be achieved.** The bars represent response averages of the given aspects.

### Actual and ideal time spent working from home

We also asked the researchers how much of their work time they spent working from home in the past, and how much it would be ideal for them to work from home in the future concerning both research efficiency and well-being. Fig 3 shows the distribution of percentages of time working from home in the past and in an ideal future. Comparing these values for each researcher, we found that 66% of them want to work more from home in the future than they did before the lockdown, whereas 16% of them want to work less from home, and 18% of them want to spend the same percentage of their work time at home in the future as before. (These latter calculations were not preregistered).

### Feasibility of working more from home

Taken all their other duties (education, administration, etc.) and provided circumstances at home (infrastructure, level of disturbance), of researchers who would like to work more from home in the future (n = 461), 86% think that it would be possible to do so. Even among those who have teaching duties at work (n = 376), 84% think that more working from home would be ideal and possible.

### Discussion

Researchers' work and life have radically changed in recent times. The flexibility allowed by the mobilization of technology and the continuous access to the internet disintegrated the traditional work-life boundary. Where, when, and how we work depends more and more on our own arrangements. The recent pandemic only highlighted an already existing task: researchers'

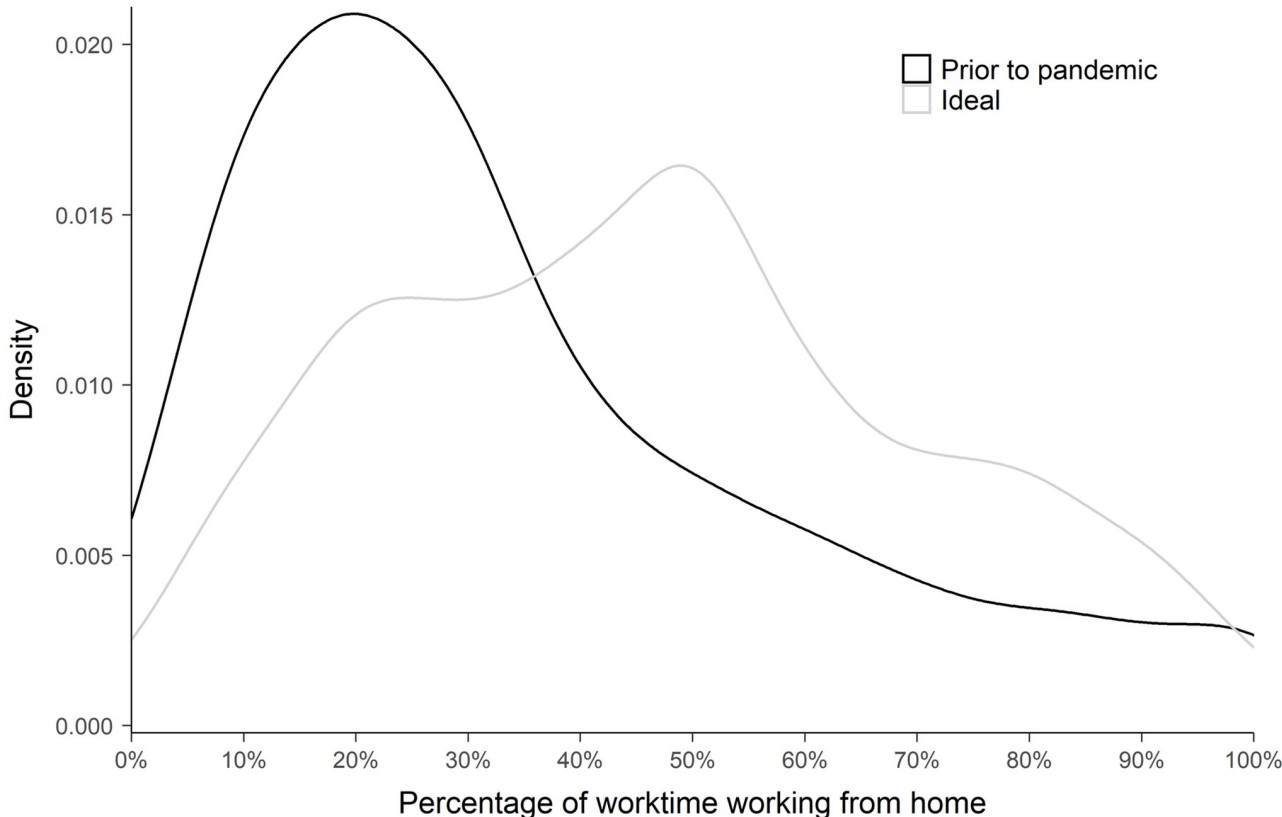

**Fig 3. The density distributions of the responses (*N* = 704) when asked how much of their worktime they worked from home before the pandemic lockdown and how much they would find ideal to work from home in the future.**

worklife has to be redefined. The key challenge in a new work-life model is to find strategies to balance the demands of work and personal life. As a first step, the present paper explored how working from home affects researchers' efficiency and well-being.

Our results showed that while the pandemic-related lockdown decreased the work efficiency for almost half of the researchers (47%), around a quarter (23%) of them experienced that they were more efficient during this time compared to the time before. Based on personal experience, 70% of the researchers think that after the lockdown they would be similarly (41%) or more efficient (29%) than before if they could spend more of their work-time at home. The remaining 30% thought that after the lockdown their work efficiency would decrease if they worked from home, which is noticeably lower than the 47% who claimed the same for the lockdown period. From these values we speculate that some of the obstacles of their work efficiency were specific to the pandemic lockdown. Such obstacles could have been the need to learn new methods to teach online [44] or the trouble adapting to the new lifestyle [45]. Furthermore, we found that working from the office and working from home support different aspects of research. Not surprisingly, activities that involve colleagues or team members are better bound to the office, but tasks that need focused attention, such as working on the manuscript or analyzing the data are better achieved from home.

A central motivation of our study was to explore what proportion of their worktime researchers would find ideal to work from home, concerning both research efficiency and work-life balance. Two thirds of the researchers indicated that it would be better to work more from home in the future. It seemed that sharing work somewhat equally between the two

venues is the most preferred arrangement. A great majority (86%) of those who would like to work more from home in the future, think that it would be possible to do so. As a conclusion, both the work and non-work life of researchers would take benefits should more WFH be allowed and neither workplace duties, nor their domestic circumstances are limits of such a change. That researchers have a preference to work more from home, might be due to the fact that they are more and more pressured by their work. Finishing manuscripts, and reading literature is easier to find time for when working from home.

A main message of the results of our present survey is that although almost half of the respondents reported reduced work efficiency during the lockdown, the majority of them would prefer the current remote work setting to some extent in the future. It is important to stress, however, that working from home is not equally advantageous for researchers. Several external and personal factors must play a role in researchers' work efficiency and work-life balance. In this analysis, we concentrated only on family status, but further dedicated studies will be required to gain a deeper understanding of the complex interaction of professional, institutional, personal, and domestic factors in this matter. While our study could only initiate the exploration of academics' WFH benefits and challenges, we can already discuss a few relevant aspects regarding the work-life interface.

Our data show that researchers who live with dependent children can exploit the advantages of working from home less than those who do not have childcare duties, irrespective of the pandemic lockdown. Looking after children is clearly a main source of people's task overload and, as a result, work-family conflict [46, 47]. As an implication, employers should pay special respect to employees' childcare situations when defining work arrangements. It should be clear, however, that other caring responsibilities should also be respected such as looking after elderly or disabled relatives [48]. Furthermore, to avoid equating non-work life with family-life, a broader diversity of life circumstances, such as those who live alone, should be taken into consideration [49].

It seems likely that after the pandemic significantly more work will be supplied from home [50]. The more of the researchers' work will be done from home in the future, the greater the challenge will grow to integrate their work and non-work life. The extensive research on work-life conflict, should help us examine the issue and to develop coping strategies applicable for academics' life. The Boundary Theory [26, 51, 52] proved to be a useful framework to understand the work-home interface. According to this theory, individuals utilize different tactics to create and maintain an ideal level of work-home segmentation. These boundaries often serve as "mental fences" to simplify the environment into domains, such as work or home, to help us attend our roles, such as being an employee or a parent. These boundaries are more or less permeable, depending on how much the individual attending one role can be influenced by another role. Individuals differ in the degree to which they prefer and are able to segment their roles, but each boundary crossing requires a cognitive "leap" between these categories [53]. The source of conflict is the demands of the different roles and responsibilities competing for one's physical and mental resources. Working from home can easily blur the boundary between work and non-work domains. The conflict caused by the intrusion of the home world to one's work time, just as well the intrusion of work tasks to one's personal life are definite sources of weakened ability to concentrate on one's tasks [54], exhaustion [55], and negative job satisfaction [56].

What can researchers do to mitigate this challenge? Various tactics have been identified for controlling one's borders between work and non-work. One can separate the two domains by temporal, physical, behavioral, and communicative segmentation [26]. Professionals often have preferences and self-developed tactics for boundary management. People who prefer tighter boundary management apply strong segmentation between work and home [57, 58]. For instance, they don't do domestic tasks in worktime (temporal segmentation), close their

door when working from home (physical segmentation), don't read work emails at weekends (behavioral segmentation), or negotiate strict boundary rules with family members (communicative segmentation). People on the other on one side of the segmentation-integration continuum, might not mind, or cannot avoid, ad-hoc boundary-crossings and integrate the two domains by letting private space and time be mixed with their work.

Researchers, just like other workers, need to develop new arrangements and skills to cope with the disintegration of the traditional work-life boundaries. To know how research and education institutes could best support this change would require a comprehensive exploration of the factors in researchers' WFH life. There is probably no one-size-fits-all approach to promote employees' efficiency and well-being. Life circumstances often limit how much control people can have over their work-life boundaries when working from home [59]. Our results strongly indicate that some can boost work efficiency and wellbeing when working from home, others need external solutions, such as the office, to provide boundaries between their life domains. Until we gain comprehensive insight about the topic, individuals are probably the best judges of their own situation and of what arrangements may be beneficial for them in different times [60]. The more autonomy the employers provide to researchers in distributing their work between the office and home (while not lowering their expectations), the more they let them optimize this arrangement to their circumstances.

Our study has several limitations: to investigate how factors such as research domain, seniority, or geographic location contribute to WFH efficiency and well-being would have needed a much greater sample. Moreover, the country of residence of the respondents was not collected in our survey and this factor could potentially alter the perception of WFH due to differing social and infrastructural factors. Whereas the world-wide lockdown has provided a general experience to WFH to academics, the special circumstances just as well biased their judgment of the arrangement. With this exploratory research, we could only scratch the surface of the topic, the reader can probably generate a number of testable hypotheses that would be relevant to the topic but we could not analyze in this exploration.

Newton working in lockdown became the idealized image of the home-working scientist. Unquestionably, he was a genius, but his success probably needed a fortunate work-life boundary. Should he had noisy neighbours, or taunting domestic duties, he might have achieved much less while working from home. With this paper, we aim to draw attention to how WFH is becoming a major element of researchers' life and that we have to be prepared for this change. We hope that personal experience or the topic's relevance to the future of science will invite researchers to continue this work.

## Supporting information

**S1 File.**
(DOCX)

## Acknowledgments

We would like to thank Szonja Horvath, Matyas Sarudi, and Zsuzsa Szekely for their help with reviewing the free text responses.

## Author Contributions

**Conceptualization:** Balazs Aczel, Tanja van der Lippe, Barnabas Szaszi.

**Data curation:** Marton Kovacs.

**Formal analysis:** Marton Kovacs.

**Investigation:** Balazs Aczel, Marton Kovacs, Barnabas Szaszi.

**Methodology:** Balazs Aczel, Marton Kovacs, Tanja van der Lippe, Barnabas Szaszi.

**Project administration:** Balazs Aczel.

**Resources:** Balazs Aczel, Marton Kovacs.

**Software:** Balazs Aczel.

**Supervision:** Tanja van der Lippe, Barnabas Szaszi.

**Validation:** Balazs Aczel, Marton Kovacs, Barnabas Szaszi.

**Visualization:** Marton Kovacs.

**Writing – original draft:** Balazs Aczel, Marton Kovacs, Barnabas Szaszi.

**Writing – review & editing:** Balazs Aczel, Marton Kovacs, Tanja van der Lippe, Barnabas Szaszi.

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
