## [Decision Letter · Decision Letter 0]

8 Feb 2021

PONE-D-20-30010

Researchers working from home: Benefits and challenges

PLOS ONE

Dear Dr. Aczel,

Thank you for submitting your manuscript to PLOS ONE. After careful consideration, we feel that it has merit but does not fully meet PLOS ONE’s publication criteria as it currently stands. Therefore, we invite you to submit a revised version of the manuscript that addresses the points raised during the review process.

We look forward to receiving your revised manuscript.

Kind regards,

Johnson Chun-Sing Cheung, D.S.W.

Academic Editor

PLOS ONE

2. Please ensure that the methods, including the sampling strategy, are detailed enough to enable replication and peer review using the information provided in the main body of the manuscript. Please move information from the supporting materials as necessary.

**Comments to the Author**

Reviewer #1: PONE-D-20-30010

Title: Researchers working from home: Benefits and challenges

Reviewer’s article summary: This manuscript provides results from a survey on work-life balance among academics who switched to remote work-from-home during the Covid-19 pandemic. I believe the article contributes insight on both the work-life balance among academics and how researchers have experienced their work during the pandemic, and will be of interest to the PloS One audience. Below, please see suggestions for improving the manuscript.

Abstract: Please include a brief statement about methodology, including sample size of the survey population, how the survey was conducted (convenience sample? Recruitment strategy?).

Introduction: The authors questions, “Is the relation between working from home and productivity influenced by personal and professional factors?” This question seems like a non-starter – how could working from home not be influenced by personal and professional factors? Advise revising this question to better focus your key arguments (i.e. what personal and professional factors most influence the productivity of working from home?).

“just as well increased autonomy over time use” – awkward sentence; please revise to clarify.

“physical and social distance to teal members” – do you mean team members?

Table 1 – please refer to the table in the text to guide the audience to this comparison of pros/cons in context of the introduction. It may also better position this manuscript within the literature to include more details from the studies that list these pros/cons (i.e. include the % of people who have reported each of the pros/cons within the table itself, and include a reference to the study where each % was derived).

Reference to Snizek in the 80’s – the benefit of including this quote is questionable; it would be more helpful to include more recent literature on this point since generational changes have perhaps changed this experience.

“just as well high levels of work productivity and satisfaction” – awkward sentence, please revise for clarity.

Materials and Methods: Please provide the study number for IRB approval.

The authors do include links to their study procedure, but it would be helpful for a more complete overview of the procedure within the manuscript so the audience can more easily ascertain the methodology employed. In comparison, the “Materials” section provides intricate detail that may not be necessary (in this reviewer’s opinion, it would be more efficient to simply list the types of questions asked—i.e. “Survey questions asked participants to report on changes that occurred in relation to research work efficiency, comparison of home to office work, amount of time spent…”(etc. or something of this nature)–with a link to the actual survey instrument).

There is no section or statement regarding data analysis. Please describe your analytical procedure (descriptive statistics, any regressions?) and software used for analysis.

Results – Recommend providing a demographics table in the manuscript that displays sample size and % for the information described in the “background information” section. Please include data about the countries where respondents live, if available; if not available, please include a statement regarding residence in the Methods section (i.e. was the sample all within a single country?).

Figures – please include sample size (n = ) in the figure titles.

Discussion

” From these values we can assume that some of the obstacles of their work were specific to the pandemic lockdown and not directly to working from home” – please explain and clarify.

“…seems to be a generally wanted and beneficial model of work” – this statement seems to ignore the result that nearly half of respondents reported being less efficient during the pandemic. Recommend revising this statement, and including a summary that the results indicate although almost half of the respondents reported reduced work efficiency, they would prefer the current remote work setting to some extent in the future. May also be useful to note that the implications of this require further investigation – what is it about this new work situation that people prefer? What amount of time did people previously spend in commute that they now can use for other tasks or personal interests? What other factors have changed that make the current situation more preferred?

References

#5 – incomplete reference

There are several references that are now quite old (1987, 1996, 1999, 2001, 2002, 2003, 2006, 2007, 2008, 2009…) – Recommend reviewing these carefully to ensure that there is not more recent literature that would shed better light on the subject.

Figure 1 – recommend revising the X axis to show sample size, and the bar labels to show % to increase clarity of results.

---

## [Author Response · Author response to Decision Letter 0]

17 Feb 2021

Dear Dr. Johnson Cheung,

We are happy to submit a revised version of our manuscript to PLOS One.

We would like to thank you and the reviewer for their comments and suggestions.

Below, you can find the detailed responses to all comments in bold.

Kind regards,

Balazs Aczel, on behalf of all co-authors

Comments to the Author

Reviewer #1

Comment 1

Abstract: Please include a brief statement about methodology, including sample size of the survey population, how the survey was conducted (convenience sample? Recruitment strategy?).

Reply 1

We have added these aspects to the Abstract.

Comment 2

Introduction: The authors questions, “Is the relation between working from home and productivity influenced by personal and professional factors?” This question seems like a non-starter – how could working from home not be influenced by personal and professional factors? Advise revising this question to better focus your key arguments (i.e. what personal and professional factors most influence the productivity of working from home?).

Reply 2

We agree with the reviewer and changed that question as suggested.

Comment 3

“just as well increased autonomy over time use” – awkward sentence; please revise to clarify.

Reply 3

Fixed.

Comment 4

“physical and social distance to teal members” – do you mean team members?

Reply 4

Fixed.

Comment 5

Table 1 – please refer to the table in the text to guide the audience to this comparison of pros/cons in context of the introduction. It may also better position this manuscript within the literature to include more details from the studies that list these pros/cons (i.e. include the % of people who have reported each of the pros/cons within the table itself, and include a reference to the study where each % was derived).

Reply 5

Table 1 is referred to in the text, just above the table. After due consideration of this suggestion, we judged that three paragraphs about the pros/cons provide sufficient details on the given topic. We found no sound way to merge the empirical reports of the referred studies to provide overall percentages of people reporting each pros/cons. 

Comment 6

Reference to Snizek in the 80’s – the benefit of including this quote is questionable; it would be more helpful to include more recent literature on this point since generational changes have perhaps changed this experience.

Reply 6

The old Snizek reference serves as an indicator that academics have already experienced some of the drawbacks of working from home at the start of the popularity of personal computers. We have now extended our Introduction with more studies from the recent literature, especially with those conducted during the pandemic.

Comment 7

“just as well high levels of work productivity and satisfaction” – awkward sentence, please revise for clarity.

Reply 7

Fixed.

Comment 8

Materials and Methods: Please provide the study number for IRB approval.

Reply 8

Added.

Comment 9

The authors do include links to their study procedure, but it would be helpful for a more complete overview of the procedure within the manuscript so the audience can more easily ascertain the methodology employed. In comparison, the “Materials” section provides intricate detail that may not be necessary (in this reviewer’s opinion, it would be more efficient to simply list the types of questions asked—i.e. “Survey questions asked participants to report on changes that occurred in relation to research work efficiency, comparison of home to office work, amount of time spent…”(etc. or something of this nature)–with a link to the actual survey instrument).

Reply 9

We have now placed the Procedure section before the Materials section. At the beginning of the Materials section, we provide a link to the original content of our Qualtrics survey. This file contains the wording of the items and the display logic of the questions. We would also prefer to keep the detailed description of the survey items in the manuscript as most of the items were developed by the authors for the study. Should the Editor prefer that, we could move the Materials section to the Supporting Information and leave just the link to the exact survey questions in the manuscript.

Comment 10

There is no section or statement regarding data analysis. Please describe your analytical procedure (descriptive statistics, any regressions?) and software used for analysis.

Reply 10

Now, we state in the Data preprocessing and Analyses section that we used the R statistical software for the analyses and that we report only descriptive statistical results in this study.

Comment 11

Results – Recommend providing a demographics table in the manuscript that displays sample size and % for the information described in the “background information” section. Please include data about the countries where respondents live, if available; if not available, please include a statement regarding residence in the Methods section (i.e. was the sample all within a single country?).

Reply 11

The table with the sample size and proportions for all the levels of all the survey items is provided in the Supplementary Materials. However, as the whole table is more than 4 pages long, we think that by including the table in the main text we would corrupt the readability of the manuscript.

Now, we state in the Sampling section that the country of residence of the respondents is not known.

Comment 12

Figures – please include sample size (n = ) in the figure titles.

Reply 12

The sample sizes are now included in the figure titles.

Comment 13

Discussion

” From these values we can assume that some of the obstacles of their work were specific to the pandemic lockdown and not directly to working from home” – please explain and clarify.

Reply 13

We would like to thank the reviewer for pointing out the vagueness of this section. We rephrased the sentence and added one more sentence to the section to clarify our point.

Comment 14

“…seems to be a generally wanted and beneficial model of work” – this statement seems to ignore the result that nearly half of respondents reported being less efficient during the pandemic. Recommend revising this statement, and including a summary that the results indicate although almost half of the respondents reported reduced work efficiency, they would prefer the current remote work setting to some extent in the future. May also be useful to note that the implications of this require further investigation – what is it about this new work situation that people prefer? What amount of time did people previously spend in commute that they now can use for other tasks or personal interests? What other factors have changed that make the current situation more preferred?

Reply 14

We have now updated this sentence incorporating the reviewer’s suggestion. The updated paragraph is on page 16.

Comment 15

References

#5 – incomplete reference

There are several references that are now quite old (1987, 1996, 1999, 2001, 2002, 2003, 2006, 2007, 2008, 2009…) – Recommend reviewing these carefully to ensure that there is not more recent literature that would shed better light on the subject.

Reply 15

We fixed the incomplete reference. 

We agree that some of our references are from the ‘80s or ‘90s, yet they are still good sources of our claims (e.g., how researchers found working from home when personal computers started or that setting up a home office comes with physical and infrastructural demands). Nevertheless, we have added more recent studies to our references, especially from the relevant literature that has been published since our initial submission 5 months ago:

Johnson N, Veletsianos G, Seaman J. US Faculty and Administrators’ Experiences and Approaches in the Early Weeks of the COVID-19 Pandemic. Online Learn. 2020;24(2):6–21.

Barrero JM, Bloom N, Davis SJ. Why Working From Home Will Stick. Univ Chic Becker Friedman Inst Econ Work Pap. 2020;(2020–174).

Korbel JO, Stegle O. Effects of the COVID-19 pandemic on life scientists. Genome Biol. 2020;21(113).

Ghaffarizadeh SA, Ghaffarizadeh SA, Behbahani AH, Mehdizadeh M, Olechowski A. Life and work of researchers trapped in the COVID-19 pandemic vicious cycle. bioRxiv. 2021;

Comment 16

Figure 1 – recommend revising the X axis to show sample size, and the bar labels to show % to increase clarity of results.

Reply 16

Thank you for the recommendation. We have now modified this figure.

---

## [Decision Letter · Decision Letter 1]

23 Feb 2021

PONE-D-20-30010R1

Researchers working from home: Benefits and challenges

PLOS ONE

Dear Dr. Aczel,

Thank you for submitting your manuscript to PLOS ONE. After careful consideration, we feel that it has merit but does not fully meet PLOS ONE’s publication criteria as it currently stands. Therefore, we invite you to submit a revised version of the manuscript that addresses the points raised during the review process.

We look forward to receiving your revised manuscript.

Kind regards,

Johnson Chun-Sing Cheung, D.S.W.

Academic Editor

PLOS ONE

Reviewer #1:

PONE-D-20-30010

Title: Researchers working from home: Benefits and challenges

Reviewer’s response to revisions: Overall, the authors have revised the manuscript to increase clarity and improve understanding of the contributions that this research provides regarding the future outlook for academics working from home. I have a few minor comments:

Limitations: This revised document brings to light the fact that 1) we do not know how the transition to working from home differs between countries since country was not a survey question (which could differ significantly given a number of social and technological/infrastructure factors), and 2) since the analysis only included descriptive statistics there is great potential in learning more from this dataset – and it is wonderful that the dataset will be publicly available. I do recommend adding a statement on limitations, both because it is a best practice, and because it shows that the authors have been thoughtful about the limits of their current analysis.

Results – Recommend providing a demographics table in the manuscript that displays sample size and % for the information described in the “background information” section. I appreciate the authors’ response to this request, but suggest that as a standard practice a shortened version of the key demographics could be provided in a table within the text, and the remainder of the demographics table could be in the supplemental material (having these results within the table is standard in my field since it provides the background information necessary for academics to easily understand the full scope of the results). In response to the question of length, I would suggest that the paragraph that lists the % of respondents who were male/female, etc. could be shortened and simply refer to the table instead.

Figure 1 – recommend revising the X axis to show sample size, and the bar labels to show % to increase clarity of results. The authors responded that this change was made in the revision, but I could not find the updated figure in the revised document.

---

## [Author Response · Author response to Decision Letter 1]

4 Mar 2021

Dear Dr. Johnson Cheung,

We are happy to submit a revised version of our manuscript to PLOS One.

We would like to thank you and the reviewer for their comments and suggestions.

Below, you can find the detailed responses to all comments in bold.

Kind regards,

Balazs Aczel, on behalf of all co-authors

Comments to the Author

Reviewer #1

Comment 1

Overall, the authors have revised the manuscript to increase clarity and improve

understanding of the contributions that this research provides regarding the future outlook for

academics working from home. I have a few minor comments:

Limitations: This revised document brings to light the fact that 1) we do not know how the

transition to working from home differs between countries since country was not a survey

question (which could differ significantly given a number of social and

technological/infrastructure factors), and 2) since the analysis only included descriptive

statistics there is great potential in learning more from this dataset – and it is wonderful that

the dataset will be publicly available. I do recommend adding a statement on limitations, both

because it is a best practice, and because it shows that the authors have been thoughtful

about the limits of their current analysis.

Reply 1

We have now included a statement of limitations regarding the missing information

on country of residence and made it more clear in the limitations section that the

present study was only exploratory.

Comment 2

Results – Recommend providing a demographics table in the manuscript that displays

sample size and % for the information described in the “background information” section. I

appreciate the authors’ response to this request, but suggest that as a standard practice a

shortened version of the key demographics could be provided in a table within the text, and

the remainder of the demographics table could be in the supplemental material (having

these results within the table is standard in my field since it provides the background

information necessary for academics to easily understand the full scope of the results). In

response to the question of length, I would suggest that the paragraph that lists the % of

respondents who were male/female, etc. could be shortened and simply refer to the table

instead.

Reply 2

We have now included the key demographics as a table (Table 2) in the manuscript in

addition to the full summary of all the responses in the Supplementary information.

Comment 3

Figure 1 – recommend revising the X axis to show sample size, and the bar labels to show

% to increase clarity of results. The authors responded that this change was made in the

revision, but I could not find the updated figure in the revised document.

Reply 3

We made sure that all the figures are updated and uploaded with this submission

---

## [Editor Report · Decision Letter 2]

12 Mar 2021

Researchers working from home: Benefits and challenges

PONE-D-20-30010R2

Dear Dr. Aczel,

We’re pleased to inform you that your manuscript has been judged scientifically suitable for publication and will be formally accepted for publication once it meets all outstanding technical requirements.

Kind regards,

Johnson Chun-Sing Cheung, D.S.W.

Academic Editor

PLOS ONE

---

## [Editor Report · Acceptance letter]

16 Mar 2021

PONE-D-20-30010R2 

Researchers working from home: Benefits and challenges 

Dear Dr. Aczel:

I'm pleased to inform you that your manuscript has been deemed suitable for publication in PLOS ONE. Congratulations! Your manuscript is now with our production department. 

Kind regards, 

on behalf of

Dr. Johnson Chun-Sing Cheung 

Academic Editor

PLOS ONE